# Comparison of Ultra-Short Race Pace and High-Intensity Interval Training in Age Group Competitive Swimmers

**DOI:** 10.3390/sports11090186

**Published:** 2023-09-18

**Authors:** Konstantinos Papadimitriou, Athanasios Kabasakalis, Anastasios Papadopoulos, Georgios Mavridis, Georgios Tsalis

**Affiliations:** 1Faculty of Sport and Rehabilitation Sciences, Metropolitan College, University of East London, 546 24 Thessaloniki, Greece; 2Laboratory of Evaluation of Human Biological Performance, School of Physical Education and Sport Science, Aristotle University of Thessaloniki, 570 01 Thessaloniki, Greece; kabasakalis@phed.auth.gr; 3School of Physical Education and Sport Science, Democritus University of Thrace, 691 00 Komotini, Greece; swimtechniq@gmail.com; 4School of Physical Education and Sport Science, Aristotle University of Thessaloniki, 570 01 Thessaloniki, Greece; mauridisg@hotmail.com; 5School of Physical Education and Sport Science, Aristotle University of Thessaloniki, 621 22 Serres, Greece; geosta@gmail.com

**Keywords:** freestyle, blood lactate, glucose, heart rate, swimming efficiency

## Abstract

The aim of this study was tο examine the acute responses to an Ultra-Short Race Pace Training (USRPT) and a High-Intensity Interval Training (HIIT), both oriented for the event of 100 m freestyle. Eighteen national-level swimmers (8 boys, 10 girls) aged 13.5 ± 0.1 years, with 8.0 ± 0.5 years of experience participated in the study. All participants completed a USRPT and a HIIT protocol consisting of 2 × 10 × 25 m (USRPT1 & USRPT2) and 5 × 50 m. Significantly higher swimming velocity (SV) were achieved in USRPT compared to HIIT (*p* < 0.001), while significantly lower distance per stroke (DPS) and stroke index (SI) were obtained (*p* = 0.007 and *p* < 0.001). Also, significantly lower blood lactate and glucose (BL & BG) concentrations were found after USRPT (*p* ≤ 0.001 and *p* = 0.037). Heart rate (HR) and rate of perceived exertion (RPE) were significantly lower after USRPT than HIIT (*p* < 0.001 and *p* = 0.015). According to the results, an USRPT swimming set consisting of 20 × 25 m at a 100 m pace seems to induce more specific responses in kinematic characteristics, biomarkers, HR and RPE compared to a 5 × 50 m HIIT set.

## 1. Introduction

Swimming coaches use a variety of high-intensity interval training (HIIT) sets to increase and even evaluate swimmers’ performance. These kinds of set-tests are usually implemented in swimming training systematically. However, their frequent use may also be avoided by coaches to reduce the monotony in swimmers’ psychology [1] and the staleness of performance [2,3]. Over the last decade, a training method called Ultra-Short Race Pace Training (USRPT), which familiarizes the swimmers with the pace of a swimming event, has appeared as an efficient way for performance enhancement and indication [2,3].

USRPT’s conceptualization mainly originated in the research of Astrand et al. [4] and Beidaris et al. [5], where a brief period of work (sets of 25 and 50 m) with a rest interval close to 20 s was considered beneficial for performance. Thus, a typical USRPT session is usually formed of short, distanced bouts (15–100 m), with brief rest periods (15–25 s) and rather high volumes (up to 5–10 × the distance of the targeted event) and performed at the pace of the targeted event, thus, with submaximal to maximal intensity. A traditional USRPT set is 20 × 50 m at the pace of 200 m with 20 s intervals between them [3]. However, there is a lack of evidence regarding the physiological responses and performance effects of this type of training. Nugent et al. [6] stated that there are no clear USRPT protocols in the literature, but mostly variations of HIIT.

HIIT is used by coaches as an effective way for swimmers’ improvement [7,8,9,10] in contrast, or supplementarily, to low or moderate intensity continuous training sets which have been found to induce fewer physiological adaptions than HIIT [10,11]. Therefore, there are many studies in which the effects of HIΙT in swimming have been studied [10,12,13,14].

According to the American College of Sports Medicine [15] the duration of a usual set of HIIT set varies from five seconds to eight minutes, at an intensity greater than 80% of maximal heart rate (HR) or VO_2_max and work-to-rest ratio of 1:2 to 1:4. In swimming, standardized HIIT sets, in adult and puberty swimmers, is four to ten repetitions of 50 m with maximal intensity and work-to-rest ratio near to 1:4 [7,8,9,16]. For example, in studies implementing swimming, intensity was indeed high with blood lactate (BL) concentrations reaching 14.1 ± 3.1 mM/L^−1^ after a 4 × 50 m maximal freestyle with a rest ratio of 1:4 [17] and rate of perceived exertion (RPE) being between 6–10 after 2 × 4 × 17.5 m freestyle and freestyle kick with work-to-rest ratio 1:5 [13]. Therefore, the fundamental differences between HIIT and USRPT sets are the interval (1:4 based on the total duration vs. 20 s after each bout) and the volume (200–500 m vs. 5–10 times the targeted event) [3,10,12,13,14]. To conclude, although the swimmers seem to have benefited from a HIIT intervention [7,8,9,16]. Rushall [3] regards the USRPT as a superior method for improving swimmers’ performance, referring to less stress on the body while certain physiological requirements are met.

In support of that notion, Cuenca-Fernández et al. [18], found that a training set of 20 × 50 m, that resembled USRPT, induced lower BL than a set of 10 × 100 m. Additionally, after the first set of USRPT, the subsequent performance declined, while the stroke rate (SR) increased. Williamson et al. [19] studied a USRPT set which consisted of 20 × 25 m with a 35 s interval at 100 m freestyle pace. According to the results, the USRPT protocol increased BL up to 13.6 ± 3.1 mM/L^−1^, RPE from 8.0 ± 1.6 to 18.0 ± 1.6 and HR up to 188 ± 9 beats per minute (bpm). Therefore, their conclusion was that USRPT is an anaerobic style of training and can be used by coaches for the improvement of a mostly anaerobic event, such as 100 m. Furthermore, in a case report where USRPT was used for the preparation of a middle-distance swimmer for 11 weeks, it appeared to be successful for performance improvement in the events of 200 and 400 m freestyle [2].

According to Rushall [3], the strengths of the USRPT method are found in the swimmers’ ability to train (1) simulating a race, (2) in actual fatigue condition, (3) with the lowest burden of BL [19], (4) understanding the magnitude of improvement after each USRPT, (5) adapting to the event’s specific kinematic conditions (stroke rate, stroke index, stroke length, etc.) and (6) predicting the performance in a forthcoming swim meeting.

Although USRPT has gained much attention, there is a lack of data about this training method, the potential responses to it, especially in young swimmers, and the claimed superiority compared to HIIT method. Thus, the aim of the present study was to examine the acute training responses of an USRPT set on kinematic characteristics, biomarkers, HR and RPE and compare them with those to a HIIT set, both oriented for the event of 100 m freestyle, in national-level adolescent swimmers. The purpose of comparing the two training methods was to explore new variations of high-intensity training. Based on the literature, our hypothesis was that USRPT would result in lower physiological demands compared to HIIT. Regarding the kinematic analysis, our expectations were that USRPT would be a more efficient training method when compared to HIIT.

## 2. Materials and Methods

### 2.1. Subjects

A total of 18 swimmers, 8 boys and 10 girls, aged 13.5 ± 0.1 years old (Mean ± SD) took part in the study. The swimmers had an average experience of 8.0 ± 0.5 years in the sport and 2 years in the competitive category, while they were trained for 12 h covering close to 30 km per week. Additionally, their best performances in the 100 m front crawl event in a long course pool were rather similar, with an average time of 66.4 ± 2.8 s. The anthropometric characteristics as well as the seasonal best performance in the short course pool at 25, 50 and 100 m front crawl were recorded prior to the study (Table 1).

Inclusion criteria were (a) healthy participants, (b) aged between 13–14 years old, (c) females in the middle period of menstruation (d) 100 m freestyle as one of their main events, (e) daily participation in swimming training, (f) participation in swimming meetings, (g) placed, in at least one swimming event, in the top sixteen of their age group’s national championship and (h) World Aquatics (WA) points > 500. Before the measurements, all the swimmers and their parents were informed about the study’s process and safety. Then, a written consent form was signed by the parents to ensure swimmers’ participation. The study was in accordance with the Declaration of Helsinki and the design was approved by the Research Ethics Committee of the School of Physical Education and Sport Science (approval number 15/2022).

### 2.2. Design

#### 2.2.1. Pre-Test Procedures

The study included two tests, a HIIT and an USRPT one and was conducted in a randomized counterbalanced order on two separate days with 24-h of recovery before and between them. All participants were familiar with HIIT and USRPT sets from their usual workouts. The swimmers were asked to follow the same nutrition plan as well as daily schedule at these two days. Before the tests, the swimmers followed the same warm-up which included a five minute full-body dry land routine of dynamic and explosive exercises. This was followed by a 20 min warm-up with 900 m of swimming that they regularly did in training and included drills, kicking and a short high intensity set (2 × 25 m front crawl with 30 s of rest in between) for priming purposes. At the end of the warm-up procedure the swimmers had ten minutes of rest and continued with the swimming tests. The measurements were conducted in an indoor 25 m pool.

#### 2.2.2. Test Protocols

The HIIT protocol consisted of 5 × 50 m front crawl [6,7,8], at maximal intensity (≈103% of best 100 m best performance pace), with 3 min intervals [6,7,8,16]. Training sets like this are mainly used by coaches for the preparation of the 100 m front crawl event [17]. The swimmers were asked to maintain a speed close to the 50 m record in each bout. In swimming, the benefits of HIIT are most pronounced at the highest intensities and when performed in volumes ranging from 200 to 500 m [10,12,13,14].The USRPT protocol consisted of 20 × 25 m front crawl, which was conducted in two equal parts of 10 × 25 m, termed USRPT1 and USRPT2 throughout [3,19], at 100 m pace (≈100% of 100 m best performance pace), with 35-s intervals [19]. Before and during this protocol, the swimmers were asked to hold the pace of the 100 m event [3,19]. The USRPT test was done in two parts of 10 × 25 m to allow measurements at the end of the first part which was of the same volume as the HIIT test. Therefore, they were compared under equal training volume and double training volume conditions. The rest interval between the two parts was 2 min. Swimming times were recorded with the use of an electronic stopwatch (Stopwatch Selecta, W10710, Waterfly) by two experienced coaches in our research group. An example of the protocol set-up for a swimmer is shown on Table 2. In contrast to HIIT, USRPT requires additional investigation regarding factors such as volume and intensity. However, according to the existing literature, the most commonly utilized training volume is 500 m, split into 25 m segments, at an intensity similar to the target race pace [2,3].

#### 2.2.3. Kinematic Analysis

A digital video camera (SONY HDR-CX450 BLACK) was placed at a high point on one side and in the center of the pool [20], 20 m from the lanes where the swimmers performed their efforts, to record the entire 25 m distance [21,22]. Τhe recordings were analyzed for each swimmer for the calculation of distance per stroke (DPS) (distance/strokes) (m), stroke rate (SR) (time of 3 cycles/3 × 60) (strokes·min^−1^), stroke index (SI) (speed × DPS) (m^2^·s^−1^) and swimming velocity (SV) (distance/time) (m·s^−1^) [23]. The kinematic characteristics were assessed by analyzing the middle 15 m of each effort.

#### 2.2.4. Biomarkers

BL and blood glucose (BG) concentrations were measured in capillary blood from a fingertip using portable analyzers (Lactate Scout 4, EΚF Diagnostics, Cardiff, UK and GM550, Bionime, Taichung, Taiwan, respectively). In the HIIT protocol, blood sampling for BL determination was taken before the set and 1 and 3 min after the 5 × 50 m repetitions, whereas BG measurements were measured before and 1 min after the set. In the USRPT protocol, blood sampling for BL determination was done before exercise, 1 min after USRPT1 and 1 and 3 min after USRTP2, while the BG measurement was done before exercise, 1 min after USRPT1 and 1 min after USRPT2 [24].

#### 2.2.5. Heart Rate

The swimmers, during the measurements, were wearing a HR sensor (Polar OH1, Finland) on their dominant arm. The sensor monitored their HR immediately before and after the 5 × 50 m sets in the HIIT protocol, as well as immediately before and after USRPT1 and USRPT2 in the USRPT protocol.

#### 2.2.6. Rate of Perceived Exertion

RPE was measured via the revised category-ratio scale (0 to 10) before and after HIIT protocol and before USRPT protocol, after USRPT1 and after [13]. The whole measurement process is depicted in Figure 1.

### 2.3. Statistical Analysis

A priori sample size (n) calculation was completed with G*Power 3.1.9.7 (Düsseldorf, Germany, Universität Düsseldorf) on Windows [25]. Determining a medium effect size at 0.34 for two groups (males and females) and five measurements (before, during and after for USRPT, before and after for HIIT) ascertain that a sample size of 18 participants, giving 95% chance to reject the null hypothesis, will be needed to detect significant differences.

All values are shown as mean with standard deviation (±). Descriptive statistics and a test of normality (Shapiro–Wilk) for all the variables were used. Additionally, the measurements were checked for homogeneity and sphericity via Box’s M test of equality of covariance and Mauchly’s test, respectively. When sphericity was not met, Greenhouse Geisser was used to analyze the results. A three-way ANOVA (protocol × gender × measurement) with repeated measures, checking for possible within or between subjects’ effects, was utilized for data analyzation. Possible statistically significant differences were analyzed via Syntax, making pairwise comparisons between groups with Bonferroni’s post hoc test. The effect size (ES) with partial Eta square (*η*^2^) was estimated (*η*^2^*_p_*; small ≥ 0.01, medium ≥ 0.06, large ≥ 0.14) [26]. Also, correlation analysis was conducted through a bivariate Pearson’s *r* analysis between the variables. The statistical analysis was performed with the software SPSS, Version 25.0 (Armonk, NY, USA: IBM Corp). The level of significance was set at *α* = 0.05.

## 3. Results

According to the Shapiro–Wilk test, the data were normally distributed (*p* > 0.05). Thus, parametric analyses were followed for all examined variables. Also, the data of the swimmers’ kinematic characteristics (DPS, SR, SI and SV), biomarkers (BL, BG), HR, and RPE showed homogeneity (*p* > 0.05), according to Box’s test of equality of covariance test metrices.

### 3.1. Kinematic Characteristics

#### 3.1.1. Distance per Stroke (DPS)

Statistically significant differences were found between USRPT and HIIT protocols (*p* < 0.05). Specifically, the swimmers had a lower DPS during USRPT1 and USRPT2 than during the HIIT set, with a large effect size (DPS_(USRPT1)_, DPS_(USRPT2)_ vs. DPS_(HIIT)_: 1.41 ± 0.81, 1.46 ± 0.23 vs. 1.63 ± 0.37 m, *p* = 0.007 and *p* < 0.001 respectively, *η*^2^ = 0.20). On the other hand, no statistically significant difference was found between genders (*p* < 0.05).

#### 3.1.2. Stroke Rate (SR)

No statistically significant differences between USRPT and HIIT protocols were found in SR.

#### 3.1.3. Stroke Index (SI)

Statistically significant differences were found between the USRPT and HIIT protocols and between genders (*p* < 0.05). Specifically, the swimmers had a lower SI during USRPT1 and USRPT2 than during the HIIT set, with a large effect size (SI_(USRPT1)_, SI_(USRPT2)_ vs. SI_(HIIT)_: 1.82 ± 0.70, 1.78 ± 0.71 vs. 2.26 ± 0.82 m^2^·s^−1^, *p* < 0.001 in both cases, *η*^2^ = 0.60). Moreover, a statistically significant difference between genders was found only in USRPT1, with a large effect size (USRPT1, males vs. females: 2.07 ± 0.10 vs. 1.56 ± 0.93 m^2^·s^−1^, *p* = 0.02; *η*^2^ = 0.10).

#### 3.1.4. Swimming Velocity (SV)

Statistically significant differences were found in SV between USRPT and HIIT protocols as well as between genders (*p* < 0.05). Specifically, the swimmers had a faster SV during USRPT1 and USRPT2 than during the HIIT set, with a large effect size (SV_(USRPT1)_, SV_(USRPT2)_ vs. SV_(HIIT)_: 1.61 ± 0.20, 1.60 ± 0.20 vs. 1.56 ± 0.20 m s^−1^, *p* < 0.001, *η*^2^ = 0.70). A statistically significant difference between genders was found only in the USRPT set, which had a large effect size (in USRPT1, males vs. females: 1.67 ± 0.03 vs. 1.55 ± 0.02 m·s^−1^, *p* = 0.004; in USRPT2, males vs. females: 1.66 ± 0.03 vs. 1.55 ± 0.03 m·s^−1^, *p* = 0.011, *η*^2^ = 0.30) (Table 3).

### 3.2. Biomarkers

#### 3.2.1. Blood Lactate (BL)

Statistically significant differences were found between USRPT measurements and between USRPT and HIIT protocols (*p* < 0.05). Specifically, the swimmers’ maximal BL concentration was higher at USRPT2 than USRPT1 (BL_(USRPT1)_ vs. BL_(USRPT2)_: 8.7 ± 0.8 vs. 10.0 ± 0.9 mmol L^−1^, *p* = 0.007). Also, in both USRPT sets, the swimmers had lower maximal BL concentration than in the HIIT set, with a large effect size (BL_(USRPT1)_ & BL_(USRPT2)_ vs. BL_(HIIT)_: 8.7 ± 0.8 and 10.0 ± 0.9 vs. 12.9 ± 0.4 mmol L^−1^, *p* < 0.001 and *p* = 0.001, respectively, *η*^2^ = 0.90) (Figure 2). On the other hand, no statistically significant difference was found between genders (*p* < 0.05).

#### 3.2.2. Blood Glucose (BG)

Statistically significant differences were found between USRPT measurements and between USRPT and HIIT protocols (*p* < 0.05). Specifically, the swimmers’ BG concentration was higher in USRPT2 than in USRPT1 (BG_(USRPT1)_ vs. BG_(USRPT2)_: 6.0 ± 0.9 vs. 6.4 ± 1.4 mmol L^−1^, *p* = 0.038). Also, the swimmers had lower BG concentrations in USRPT1 than in the HIIT set, with a large effect size (G_(USRPT1)_ vs. G_(HIIT)_: 6.0 ± 0.9 vs. 6.4 ± 0.9 mmol L^−1^, *p* = 0.037, *η*^2^ = 0.40) (Figure 3). On the other hand, no statistically significant difference was found between genders (*p* < 0.05).

### 3.3. Heart Rate (HR)

Statistically significant differences were found between USRPT measurements, USRPT and HIIT protocols and genders (*p* < 0.05). Specifically, the swimmers had higher HR in USRPT2 than USRPT1 (HR_(USRPT1)_ vs. HR_(USRPT2)_: 173 ± 1 vs. 180 ± 0 bpm, *p* < 0.001). Also, the swimmers’ HR in USRPT1 and USRPT2 was lower than in HIIT, with a large effect size (HR_(USRPT1)_, HR_(USRPT2)_ vs. HR_(HIIT)_: 173 ± 1, 180 ± 0 vs. 196 ± 1 bpm, *p* < 0.001, *η*^2^ = 0.90). Moreover, a statistically significant difference between genders was found only in USRPT2 set, with a small effect size (males vs. females _(USRPT2)_: 181 ± 1 vs. 179 ± 1 bpm, *p* = 0.004, *η*^2^ = 0.02) (Figure 4).

### 3.4. Rate of Perceived Exertion (RPE)

Statistically significant differences were found between USRPT measurements and USRPT and HIIT protocols (*p* < 0.05). Specifically, the swimmers’ RPE was higher in USRPT2 than in USRPT1 (RPE_(USRPT1)_ vs. RPE_(USRPT2)_: 7 ± 0 vs. 8 ± 0, *p* < 0.001). Also, the swimmers’ RPE was lower in USRPT1 than in HIIT, with a large effect size (RPE_(USRPT1)_ vs. RPE_(HIIT)_: 7 ± 0 vs. 8 ± 0, *p* = 0.015, *η*^2^ = 0.30) (Figure 5). On the other hand, no statistically significant difference was found between genders (*p* > 0.05).

### 3.5. Correlations

Statistically significant correlations were observed between USRPT and HIIT protocols in SR and BG. Specifically, SR in HIIT protocol showed a statistically significant correlation with SR in USRPT1 and USRPT2 (*r*^2^ = 0.816, *p* < 0.001 and *r*^2^ = 0.829, *p* < 0.001). Also, BG in the HIIT protocol was correlated with BG in USRPT1 and USRPT2 (*r* = 0.627, *p* = 0.05 & *r* = 0.773, *p* < 0.001).

## 4. Discussion

### 4.1. Conceptualization of the Study

The present study aimed to examine the acute responses of a USRPT set on kinematic characteristics, biomarkers, HR and RPE and compare them with those to a HIIT set, both sets were oriented for the event of 100 m freestyle in national-level adolescent swimmers. Since USRPT has not been extensively studied, it was reasonable to compare it with a similar training stimulus like HIIT to distinguish them [6]. Particularly in swimming, HIIT has been studied at different volumes, intervals, swimmers’ ages, or levels [11,13,14,17] supplying abundant information for its use by the swimming coaches.

In their review, Nugent et al. [6], mentioned that there was no study, to date, that utilized USRPT with Rushall’s standard principles. A year later, USRPT was utilized in a few studies [2,18,19], showing a beneficial role of that type of training in short and middle-distance swimmers. These latter studies followed more appropriate USRPT protocols for the events of 100, 200 and 400 m respectively. However, there were still some discrepancies in these studies. Williamson et al. [19] used an USRPT protocol oriented for 100 m (20 × 25 m), as in the present study, but in order to continually measure BL (after every four bouts) gave the swimmers additional rest, in contrast to USRPT guidelines. Alternatively, Cuenca-Fernández et al. [18], who used an USRPT set oriented for 200 m, measured BL only at the end of the set, while Papadimitriou [2] did not measure BL. Therefore, apart from monitoring kinematic parameters, HR, and RPE we were prompted to evaluate biomarkers in the middle as well as in the end of the USRPT to better interpret the responses to an USRPT protocol and the possible differences to a HIIT one.

### 4.2. Kinematic Characteristics

According to the basic principles of USRPT [3], the swimmers must swim in a race-specific manner [6]. Thus, DPS, SR, SI, and SV during USRPT must be like the ones at the swimmers’ event. Compared with the HIIT protocol, the swimmers had statistically significant lower DPS, no difference in SR, lower SI and greater SV in USRPT. Regarding DPS in USRPT, the higher intensity, the shorter bouts (25 m) and the greater set volume (500 m) than HIIT, are possible factors that may contributed to lower values [6,27]. However, the long-term use of USRPT, would possibly provide an adaptation in swimming efficiency such as enhanced propulsion force and by extend higher SI [2]. Also, the difference between genders, only in USRPT, is perhaps explained in part by the different parameters of the set lie volume and intensity, however, more research is needed for safer conclusions.

Regarding SR, there was no statistically significant difference between protocols while a high correlation was found (*r*^2^ = 0.816, *p* < 0.001 and *r*^2^ = 0.829, *p* < 0.001). Having a lower DPS in USRPT protocol, one would expect SR to be higher than in HIIT. However, SR is a parameter which is mainly determined by neural activation [27]. However, no similar studies comparing SR between USRPT and HIIT currently exist. The long-term familiarization with USRPT could yield some different results on SR.

In addition, in the study from Wądrzyk et al. [28], it was observed that both fast (mean FINA points for 100 m front crawl = 557) and slow (mean FINA points for 100 m front crawl = 379) international level swimmers had similar SR (fast swimmers = 49.78 ± 2.47 vs. slower = 49.21 ± 4.40 cycles/min^−1^) with our participants. Therefore, it could be hypothesized, that in swimming protocols which are oriented to a specific event distance, in this instance 100 m, swimmers show similar SR. In that sense, the absence of a difference in SR between USRPT and HIIT protocol was a rather relevant result.

SI is also an important determinant of front crawl swimming performance, especially in young and high-level swimmers [29,30]. According to the present results, the SI showed higher values in HIIT compared to USRPT protocol. This indicates that the swimmers swam more efficiently in the HIIT protocol than in the USRPT protocol.

The total volume of the sets (500 vs. 250 m) and of each repetition (25 vs. 50 m), most likely affected the SI. Notably, Sánchez and Arellano [31] suggested that, as the swimming distance increases, the SI decreases. Thus, this hypothesis possibly fits with the volume of a training set as well. In addition, Wądrzyk et al. [28], showed that in a 100 m front crawl, swimmers had greater SI than the 50 m and this hypothesis may fit with the volume of each repetition of the training set. These findings seem contradictory; however, they do demonstrate that swimming distance most likely affects SI, but not in a particular manner. Hence, the principle of USRPT setting the volume between 5 to 10 times the total distance, with short repetitions at the pace of the targeted event [3], may not provide a specific or beneficial training stimulus for improving swimming efficiency. Regarding SV, it was expected to be greater at shorter distances [6,24], found greater velocity in a set of 8 × 25 m compared to a set of 8 × 50 m. Hence, the difference in SV between USRPT and HIIT protocols is most likely of no surprise. This difference in SV possibly also explains the shorter DPS which was observed in USRPT compared with HIIT.

According to anecdotal reports, some of the swimmers’ benefits who follow the USRPT method are relevant to the events’ swimming paces and the specific kinematic patterns [3]. However, according to the present results, the acute utilization of USRPT did not demonstrate superiority in kinematic parameters compared to the HIIT protocol, apart from the SV.

### 4.3. Biomarkers

Both USRPT and HIIT protocols induced increases in BL and BG concentration. These results are similar to the ones of Terzi et al. [17], Williamson et al. [19], and Kabasakalis et al. [24]. However, BL was lower in both USRPT1 and USRPT2 compared to HIIT, while BG, which was studied for the first time in USRPT, to our knowledge, was higher after the HIIT protocol compared to USRPT1.

According to Rushall [3], Cuenca-Fernández et al. [18], and Williamson et al. [19] USRPT is a type of training which largely relies on anaerobic metabolism. Similarly, in our study, BL exceeded 8 mmol L^−1^, implying an important contribution of anaerobic metabolism to energy demands while BG also increased, except for during the first half of the set (10 × 25 m), possibly indicating a metabolically stressful stimulus to the swimmers.

However, the maximal mean BL in USRPT was lower compared to the values found by Williamson et al. [19]. It is believed that adolescent athletes mostly utilize the aerobic energy system [12,13]. Hence, the fact that participants in the present study were adolescent while they were adult swimmers in that of Williamson et al. [19] possibly explain the difference in BL.

Additionally, Cuenca-Fernández et al. [18], who compared a USRPT with a HIIT protocol, consisting of 20 × 50 and 10 × 100 m, respectively, at a 200 m pace, found lower levels of fatigue and better recovery during USRPT. In the sense that BL indicates the degree of metabolic stress of a stimulus, at least for anaerobic metabolism, we also found that the shorter bouts at 100 m pace of USRPT induced a lower BL response. Therefore, it seems possible that the shorter distances along with the steady pace and the duration of the intervals elicit lower BL, independently of the content of the USRPT set. On the other hand, high-intensity protocols have been considered as a stronger training stimulus, according to BL and BG values [24].

Another reason that the HIIT protocol induced higher BL than USRPT is most likely that the more high-intensity work (mostly relied on the glycolytic system) before rest (where aerobic metabolism dominates) that had to be done in each repetition, as the average bout duration was 32.3 s for HIIT and 17.0 s for USRPT. This difference possibly indicates a greater glycolytic stimulus in HIIT than in USRPT, a result that Cuenca-Fernández et al. [18] found too. On the other hand, BG was similar and highly correlated between HIIT and USRPT2, implying that in high-intensity training sets that largely rely on glycolytic pathways, BG has similar rising responses [24].

Based on the present findings regarding biomarkers, the use of either USRPT or HIIT by coaches must be chosen according to the training period, the swimmers’ age and the distance of the event. Specifically, for maximal BL and stronger anaerobic stimulus in adolescent swimmers, who work for the event of 100 m, HIIT may be a better solution than USRPT. On the other hand, USRPT can be utilized when more moderate BL is required with higher SV.

### 4.4. HR and RPE

HR was lower in USRPT compared to HIIT. Moreover, HR was higher in USRPT2 than USRPT1, whereas RPE was also higher in USRPT2 than USRPT1. Therefore, the results, also considering the BL and BG results, enhance the notion that the USRPT protocol was most likely less physiologically demanding than HIIT [18]. In addition, a significant difference was found in genders in HR, only after USRPT2.

Williamson et al. [19], also monitored HR and RPE at an USRPT protocol and found comparable results with the present study. Both HR and RPE were increased after the USRPT protocol which was the same as the present study (i.e., 20 × 25 m). Also, in our study, HR from USRPT1 to USRPT2 was increased, a finding similar to that of Williamson et al. [19], where HR was increased after the seventh 25 m bout. Therefore, the HR demands increase after a specific time point of the USRPT protocol which seems to be close to the second half of the present set. On the other hand, HR values after HIIT protocol were higher than USRPT. The duration and intensity, which was maximum (≈103% of 100 m SB pace), of each HIIT bouts (50 m), might have caused these differences in HR.

Higher HR was found in males compared to females, but only after USRPT2. The literature regards that, at rest and in submaximal exercise intensity, females, because of the lower cardiac output, show higher HR than males [32]. While in maximal exercise intensity, there are cases where HR values are similar between males and females (the lack of difference in the HIIT protocol), regardless of age and the level of athletes [32,33,34,35] and others that males show higher HR values than females (the difference in the USRPT protocol) [36,37]. Thus, coaches should be aware that HR response may differ between boys and girls during high-intensity sets, but not in a consistent way.

Regarding RPE, the difference between USRPT1 and HIIT displays the higher internal load of HIIT compared to USRPT when total volume of the set is equal. On the contrary, RPE did not differ between USRPT2 and HIIT, showing that USRPT needed a double total volume of set (500 vs. 250 m, respectively) to reach the same RPE as HIIT. This pertains to Rushall’s [3] guidelines about an USRPT set volume being between five to 10 times the targeted event. However, as discussed, these volumes may affect SI, resulting in less efficient swimming. Hence, in a USRPT protocol, the volume choice may depend on the training target.

### 4.5. USRPT, a New HIIT’s Approach?

According to the results of the present study, an USRPT protocol, which consisted of 20 × 25 m on 100 m pace, can be identified as a novel approach to HIIT, inducing similar responses regarding only SR and different responses regarding DPS, SI, SV, BL, BG, HR and RPE compared to a more classic HIIT protocol which consisted of 5 × 50 m sets at maximal intensity. Coaches should be aware of the possible effects of the voluminous use of USRPT. This contrasts with Rushall’s guidelines; however, the aforementioned similar or different responses could suggest two viewpoints, one regarding the markers prioritized by coaches for the interpretation of the USRPT and HIIT effects and one regarding which markers are eventually more appropriate for the monitoring of the possible differences between these two protocols. For example, regarding the first viewpoint, coaches could choose a training protocol according to the responses of the kinematic parameters or the fluctuation of biomarkers. Regarding the second viewpoint, sport scientists could pick specific parameters for the differentiation of the responses of the two adjacent protocols.

## 5. Conclusions

According to the present findings, the acute utilization of USRPT demonstrates differences in kinematics, such as higher SV, lower DPS and SI, in biomarkers, such as lower BL and BG concentration and lower HR and RPE. To conclude, USRPT can be used by swimming coaches as a training stimulus to improve 100 m performance. However, since this suggestion is based on acute responses, it is recommended for future studies to confirm whether or not the USRPT’s possible beneficial effects on swimming performance through the evaluation of its long-term implementation.

## Figures and Tables

**Figure 1 sports-11-00186-f001:**
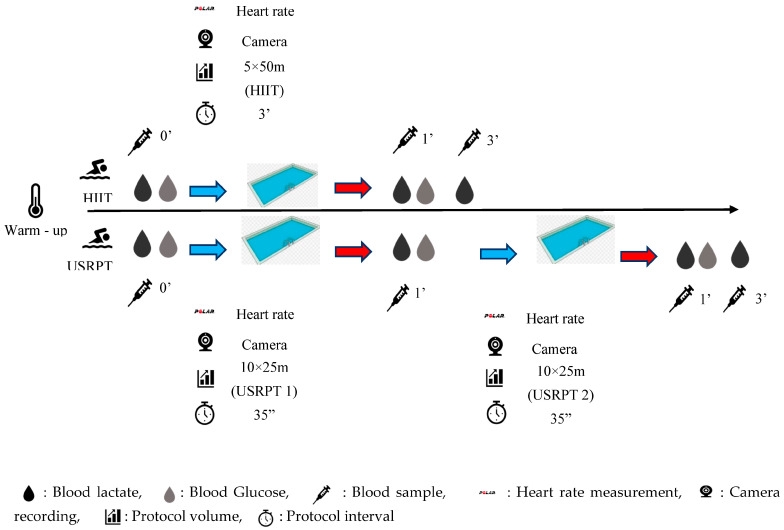
Measurement process.

**Figure 2 sports-11-00186-f002:**
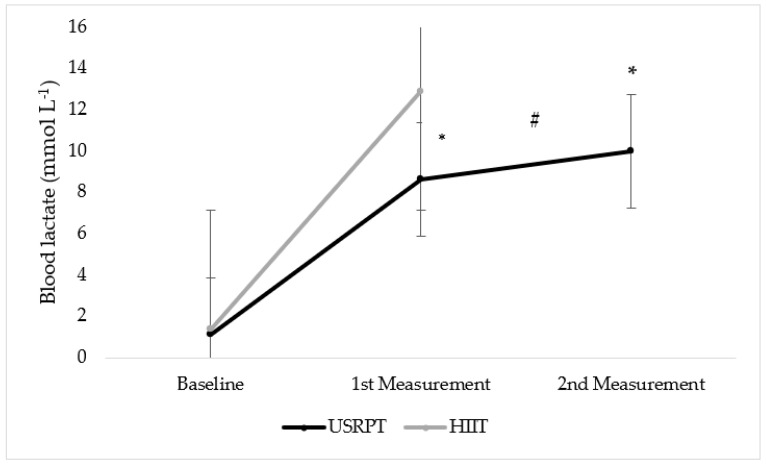
BL concentration between USRPT and HIIT protocols. * = Significant difference between protocols (*p* < 0.05), # = Significant difference between USRPT measurements (*p* < 0.05).

**Figure 3 sports-11-00186-f003:**
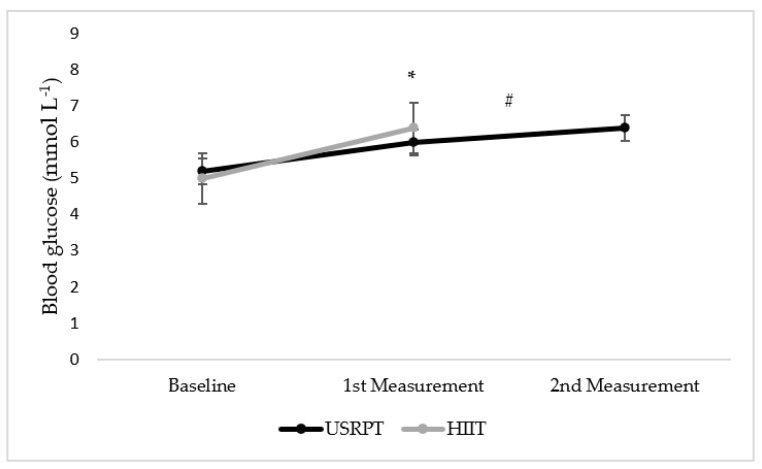
ΒG concentration between USRPT and HIIT protocols. * = Significant difference between protocols (*p* < 0.05), # = Significant difference between USRPT measurements (*p* < 0.05).

**Figure 4 sports-11-00186-f004:**
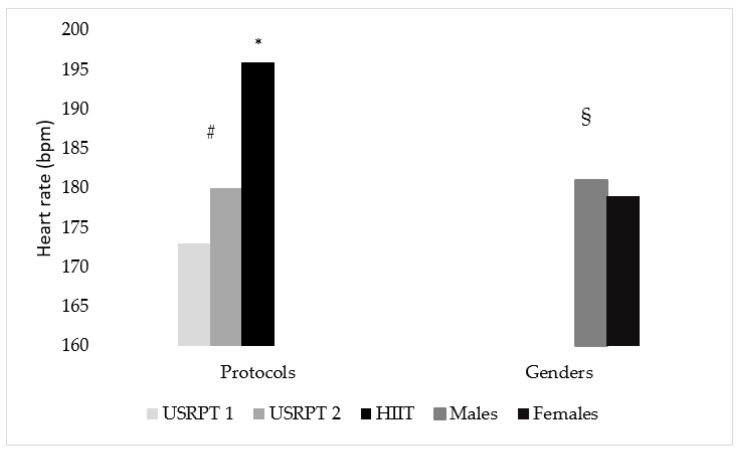
HR between USRPT and HIIT protocols. USRPT1 = 1−10 × 25 m; USRPT2 = 11−20 × 25 m; * = Significant difference between protocols (*p* < 0.05), # = Significant difference between USRPT measurements (*p* < 0.05), § = Significant difference between genders (*p* < 0.05).

**Figure 5 sports-11-00186-f005:**
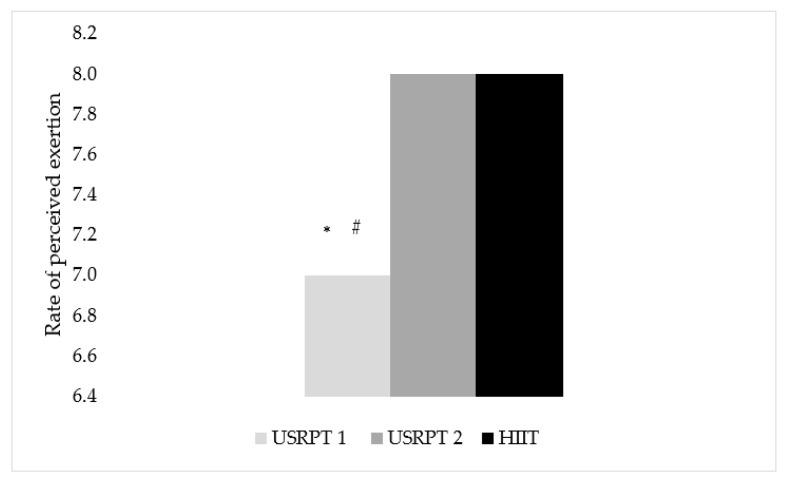
RPE between USRPT and HIIT protocols. USRPT1 = 1–10 × 25 m; USRPT2 = 11–20 × 25 m; ***** = Significant difference between protocols (*p* < 0.05), # = Significant difference between USRPT measurements (*p* < 0.05).

**Table 1 sports-11-00186-t001:** Anthropometrics and performance data of the swimmers (Mean ± SD).

Swimmers	Age (yrs)	Height (m)	Weight (kg)	BMI (kg/m^2^)	Experience (yrs)	SB 25 m (s)	SB 50 m (s)	SB 100 m (s)	WA Score (n)
Males	13.4 ± 0.2	1.67 ± 0.07	53.0 ± 3.1	18.9 ± 0.7	8.3 ± 0.9	14.1 ± 0.5	30.5 ± 0.8	65.0 ± 1.3	552 ± 170
Females	13.5 ± 0.2	1.63 ± 0.05	50.0 ± 2.4	18.8 ± 0.7	7.9 ± 0.7	14.1 ± 0.2	31.4 ± 0.4	67.6 ± 0.5	580 ± 120
Total Av.	13.5 ± 0.1	1.65 ± 0.06	51.5 ± 7.8	18.8 ± 1.9	8.0 ± 0.5	14.1 ± 0.9	31.0 ± 1.8	66.4 ± 2.8	566 ± 145

BMI = Body mass index; SB = Season best

**Table 2 sports-11-00186-t002:** Example of protocol organization.

Swimmer’s SB 100 m (s)	HIIT (s) (≈103% of 100 m SB)	USRPT (s) (100m SB/4 or ≈100% of 100 m SB)
68.0	32.5	17.0

SB: Season best; HIIT: High Intensity Interval Training; USRPT: Ultra Short Race Pace Training

**Table 3 sports-11-00186-t003:** Mean values of swimming efficiency parameters and their effect size.

	USRPT (USRPT1 & USRPT2)		HIIT	Eta Square (*η*^2^)
DPS (m)	1.41 ± 0.81	1.46 ± 0.23	1.63 ± 0.37 *	0.20
SR (strokes·min^−1^)	49.53 ± 0.78	49.41 ± 0.78	49.70 ± 0.92	0.01
SI (m^2^·s^−1^)	1.82 ± 0.70 §	1.78 ± 0.71 §	2.26 ± 0.82 *	0.60
SV (m·s^−1^)	1.61 ± 0.20 §	1.60 ± 0.20 §	1.56 ± 0.20 *	0.70

USRPT1 = 1–10 × 25 m; USRPT2 = 11–20 × 25 m; DPS = Distance Per Stroke; SR = Stroke Rate; SI = Stroke Index; SV = Swimming Velocity; * = Significant difference between protocols (*p* < 0.05); § = Significant difference between genders (*p* < 0.05).

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
