# Peer review of "Comparison of Ultra-Short Race Pace and High-Intensity Interval Training in Age Group Competitive Swimmers"

_sports, 2023, doi:10.3390/sports11090186_

Round 1
Reviewer 1 Report
Introduction
Line 37: Add another reference to the statement (has appeared as an efficient way for performance enhancement and indication [3])
Line 69: When increased stroke rate (SR)?
Materials and Methods
I recommend putting the second paragraph at the beginning of the subjects section.
How much time per week did the subjects train and how many metres did they swim?
In the design section, explain the theory behind HIIT and USRPT.
Results
Page 10 is blank, it is recommended to remove it.
Kinematic characteristics (DPS, SR and SI) should be accompanied by a table?
In Table 3 and Figure 2, 3, 4 y 5, indicate the p-value (*)
Discussion
It is recommended to start the discussion section with the aim of the study.
I think lines 349 and 350 are more in line with the methodology.
Lines 352-355: Authors are advised to get to the point.
It is recommended to summarise and be more specific in the Conceptualisation section of the study. Authors can merge both sections...
Line 376-377: “…would possibly provide an adaptation in swimming efficiency. What would be the adaptation(s)?
Line 377: Are there other studies with gender comparisons that could improve the authors' answer? There is only one reference [28]
Line 382-383: Could you provide the correlation data and compare it with any studies in the literature if there are relation.
References
Reference 22: The title of the magazine is not correct.
Author Response
Reviewer 1
Dear reviewers
We really appreciate your comments and quick review of our manuscript. Your comments helped us to improve the manuscript.
C: Comment
A: Answer
Introduction
- Line 37: Add another reference to the statement (has appeared as an efficient way for performance enhancement and indication [3])
- Done
C . Line 69: When increased stroke rate (SR)?
A . We explained the time point of the SR increment. Lines 73 – 74.
Materials and Methods
C . I recommend putting the second paragraph at the beginning of the subjects section.
A . Done
C . How much time per week did the subjects train and how many metres did they swim?
A .We included the information. Lines 101 – 103. The swimmers had an average experience of 8.0 ± 0.5 years in the sport and 2 years in the competitive category and they trained for 12 hours covering close to 30 km per week.
C . In the design section, explain the theory behind HIIT and USRPT.
A . We included some additional information about the theory behind the two training methods. Lines 137 – 139 & 149 – 153.
In swimming, the benefits of HIIT are most pronounced when performed at the highest intensities and in volumes ranging from 200 to 500 m [10,12,13,14].
In contrast to HIIT, USRPT requires additional investigation regarding factors such as volume and intensity. However, according to the existing literature, the most commonly utilized training volume is the 500 m, split into 25-m segments, at an intensity similar to the target race pace.
Results
C . Page 10 is blank, it is recommended to remove it.
A . Done
C .Kinematic characteristics (DPS, SR and SI) should be accompanied by a table?
A . It may be considered a bit redundant, but we chose to present them in a table too for a better readability and comprehension.
C . In Table 3 and Figure 2, 3, 4 y 5, indicate the p-value (*)
A . Done
Discussion
- C. It is recommended to start the discussion section with the aim of the study.
A . Done. Lines 380 – 382.
C . I think lines 349 and 350 are more in line with the methodology.
A . We appreciate your point. We deleted these sentences.
C . Lines 352-355: Authors are advised to get to the point.
A . We deleted these sentences.
C . It is recommended to summarise and be more specific in the Conceptualisation section of the study. Authors can merge both sections...
A . We deleted many sentences, keeping only the ones that we deem most relevant for the discussion.
C . Line 376-377: “…would possibly provide an adaptation in swimming efficiency. What would be the adaptation(s)?
A . We included an adaptation. Lines 417 – 418.
C . Line 377: Are there other studies with gender comparisons that could improve the authors' answer? There is only one reference [28].
A . Since supporting data for this are weak, we rephrased these lines, omitting any references. Lines 419 – 420.
C . Line 382-383: Could you provide the correlation data and compare it with any studies in the literature if there are relation.
A . Nice point. We included correlation data, but there are no similar studies examining SR between USRPT and HIIT, highlighting this research gap. Lines 427 – 428.
References
C . Reference 22: The title of the magazine is not correct.
A . Corrected

Reviewer 2 Report
Dear authors, please check my comments below (check my comments and change at will if it sounds for you)
Abstract
The abstract contains all the necessary components, giving the objective, methods, results and conclusions. At this point, the authors could give on sentence contextualizing. Nonetheless, the abstract adequately summarizes the article's main points, matching that in the body of the manuscript regarding data and terminology. Again, at this point, authors should correct 'years of training' to 'years of experience' as in the body of the text.
In my opinion, the abstract will gain readers' attention.
Introduction
The authors provide adequate context for their topic, justifying why the topic is important and timely, citing other relevant research. All the introduction is well written with an inverse pyramid form, starting with general-sounding content and then clearly stating the purpose and the hypothesis at the end.
Will the general readership of the journal find the topic meaningful?
Methods
The research design is strong and sufficiently described so that the study could be replicated by another researcher. I liked to read the inclusion criteria, referring to the level of the swimmers. That is a good point that sometimes is missing in some papers. At this point, the authors should probably change swimming speed to swimming velocity. The term velocity is the correct one. The balanced volume of the workout was a good idea. I was expecting to read methods to check on that.
At the statistics, you might want to check Abt et al (Abt, G., Boreham, C., Davison, G., Jackson, R., Nevill, A., Wallace, E., & Williams, M. (2020). Power, precision, and sample size estimation in sport and exercise science research. Journal of Sports Sciences, 38(17), 1933–1935. https://doi.org/10.1080/02640414.2020.1776002) and get some rationale for the .34 ES.
Why Seo et al. 2021 when referring to the magnitude of ES? Typically ES come after Cohen; Hopkins; Atkinson; Batterham, or other statistician.
I cannot understand the Manova with repeated measures. The protocol is not a repeated measure.
Results
The results contain all the outcome measures described in the methods. Nonetheless, I think presenting on text or with a table is more economical, and not both. Doing so, in the table, add the p-value. Do not forget to change speed to velocity. The difference between genders was given in the GLM as a gender effect ?
In the RPE result (line 323 RPE USRPT1 vs RPE USRPT2), add the ES of the difference (ex. Cohen’s d).
The correlations don't seem to add anything to the manuscript.
Comment
Well done discussing the conceptualization of the study.
The discussion is relevant, and the authors argue their findings in the context of existing research, pointing out the relevance and importance of their findings to the specific area. Make line 408 clearer to readers.
Conclusion
Check the objectives and conclude under those, along with the presented data.
Figures and Tables
The information in the tables and figures is easy to interpret, being a table detailed enough to stand independently without reference to the text (referred to before).
The table and figure information match the data in the text.
References
Recent and pertinent scientific literature is cited, and all original.
Author Response
Reviewer 2
Dear reviewers
We really appreciate your comments and quick review of our manuscript. Your comments helped us to improve the manuscript.
C: Comment
A: Answer
Abstract
C . The abstract contains all the necessary components, giving the objective, methods, results and conclusions. At this point, the authors could give on sentence contextualizing. Nonetheless, the abstract adequately summarizes the article's main points, matching that in the body of the manuscript regarding data and terminology. Again, at this point, authors should correct 'years of training' to 'years of experience' as in the body of the text.
A . We really appreciate your kind words. We modified the abstract as you suggested.
Introduction
C . Will the general readership of the journal find the topic meaningful?
A . Thank you for the question. Of course, the topic of our article specifies on a training method that has slight differences compared to HIIT. However, generally, pace training is of interest in many sports like track and field, football, basketball, etc., Coaches commonly use this type of training to improve their athletes' pace during games. As a result, we believe that the general readership will find this topic meaningful.
.
Methods
C . At this point, the authors should probably change swimming speed to swimming velocity. The term velocity is the correct one.
A . We converted “speed” into “velocity” throughout.
C . At the statistics, you might want to check Abt et al (Abt, G., Boreham, C., Davison, G., Jackson, R., Nevill, A., Wallace, E., & Williams, M. (2020). Power, precision, and sample size estimation in sport and exercise science research. Journal of Sports Sciences, 38(17), 1933–1935. https://doi.org/10.1080/02640414.2020.1776002) and get some rationale for the .34 ES.
A . Thank you for the reference. We recognize the importance of a priori G power analysis and acknowledge the potential bias that may arise. In our case, we aimed to predict the sample size based on the minimum number of swimmers, considering our inclusion and exclusion criteria as well as participants' willingness to take part. This led us to calculate a small to medium effect size (Cohen’s d test, approximately 0.34), and we determined that a minimum of 18 participants would suffice, which ultimately became our sample size.
C . Why Seo et al. 2021 when referring to the magnitude of ES? Typically ES come after Cohen; Hopkins; Atkinson; Batterham, or other statistician.
A . We agree with the comment about the usual reference by Cohen et al. We used Seo et al. because their methodology depicts the magnitude of ES, too. However, we used the most relevant reference as you suggested.
C . I cannot understand the Manova with repeated measures. The protocol is not a repeated measure.
A . We used a three-way Anova with repeated measures because we had the same sample for HIIT and USRPT conditions. Therefore, we had to compare more than two-time points (BASELINE, USRPT 1, USRPT2 and BASELINE, and HIIT).
Results
C . Nonetheless, I think presenting on text or with a table is more economical, and not both. Doing so, in the table, add the p-value. Do not forget to change speed to velocity. The difference between genders was given in the GLM as a gender effect ?
A . We agree with your comment. The distinction between the text and the table lies in the fact that in the text, we highlight only the significant differences, while in the table, we include all measurements. However, we included in the description under the table the level of significance. Also, the difference between genders was given by the GLM because GENDER was one of the three depended factors in the repeated measures analysis.
C . In the RPE result (line 323 RPE USRPT1 vs RPE USRPT2), add the ES of the difference (ex. Cohen’s d).
A .We used effect size through Eta square which is an analysis between protocols’ effects (HIIT vs USRPT).
C . The correlations don't seem to add anything to the manuscript.
A . Since, to our knowledge, our data are novel, we believe that results that to date seem not so important, may be of use when more data on the topic will (hopefully) accumulate. Thus, we would like to keep them. Moreover, we believe that the correlations between SR and BG suggest similarities between protocols regardless of the different training content. Therefore, both training variations can be used if the primary goal would be to maintain comparable SR or BG.
Comment
C . Make line 408 clearer to readers.
A . Thank you for your comment. Done. Lines 447 – 450.
Conclusion
C . Check the objectives and conclude under those, along with the presented data.
A . The conclusion was modified according to the objectives of the study. Lines 546 – 548.
